# When calculators lie: A demonstration of uncritical calculator usage among college students and factors that improve performance

Mark LaCour [1]*, Norma G. Cantú[2], Tyler Davis[1]*

1 Department of Psychological Sciences, Texas Tech University, Lubbock, Texas, United States of America,
2 Department of Psychology, University of Louisiana-Lafayette, Lafayette, Louisiana, United States of America

* mark.lacour@ttu.edu (ML); tyler.h.davis@ttu.edu (TD)

**Data Availability Statement:** All data and analysis scripts are available at https://osf.io/dme98/.

**Funding:** The authors received no specific funding for this work.

## Abstract

Calculators are often unnecessary to solve routine problems, though they are convenient for offloading cognitively effortful processes. However, errors can arise if incorrect procedures are used or when users fail to monitor the output for keystroke mistakes. To investigate the conditions under which people's attention are captured by errant calculator outputs (i.e., from incorrectly chosen procedures or keystroke errors), we programmed an onscreen calculator to "lie" by changing the answers displayed on certain problems. We measured suspicion by tracking whether users explicitly reported suspicion, overrode calculator "lies", or re-checked their calculations after a "lie" was presented. In Study 1, we manipulated the concreteness of problem presentation and calculator delay between subjects to test how these affect suspicion towards "lies" (15% added to answers). We found that numeracy had no effect on whether people opted-in or out of using the calculator but did predict whether they would become suspicious. Very few people showed suspicion overall, however. For study 2, we increased the "lies" to 120% on certain answers and included questions with "conceptual lies" shown (e.g., a negative sign that should have been positive). We again found that numeracy had no effect on calculator usage, but, along with concrete formatting, did predict suspicion behavior. This was found regardless of "lie" type. For study 3, we reproduced these effects after offering students an incentive for good performance, which did raise their accuracy across the math problems overall but did not increase suspicion behavior. We conclude that framing problems within a concrete domain and being higher in numeracy increases the likelihood of spotting errant calculator outputs, regardless of incentive.

## Introduction

Mathematics education, like with the other STEM subjects, is incredibly important because of the scientific and technological benefits it reaps for society. STEM jobs are predicted to be the

**Competing interests:** The authors have declared that no competing interests exist.

fastest growing employment sector over the next several years [1] and quantitative reasoning ability has been associated with superior performance in a variety of risky decision-making tasks in the lab as well as within real world settings [2]. It should be troubling, then, that mathematical abilities among college students appear to be on a decline [3–5].

Mathematical problem solving requires a combination of conceptual knowledge and procedural skill, which researchers believe are separate but mutually reinforcing constructs [6–7]. Routine problems mostly tap procedural skill whereas non-routine problems require a mixture of the two [8]. Routine problems also tend to require a single computation, usually performed by a calculator. The user only needs to know the correct inputs and to be able to effectively evaluate the output for possible keystroke errors or, in situations where incorrect procedures were chosen, nonsensical outputs. Most mathematical problem-solving frameworks, both prescriptive [9] and descriptive [10], include a review or verification phase corresponding to this evaluation process. Studies have shown, however, that students almost never "ballpark" answers before calculating [11] and tend not to deliberately evaluate answers afterwards [12–13]. Because of the widespread use of calculators, it is prohibitive (or impossible) to run a controlled experiment to answer whether they are the cause of the decline in mathematics abilities. However, it is possible to examine the mechanisms that influence whether people notice nonsensical outputs, which may provide a proxy for how such declines may arise from overreliance on calculators.

The purpose of the present study is to investigate conditions under which students will notice errant calculator outputs. There are two general routes to evaluating candidate answers arrived at via calculator: deliberately or automatically. The former consists of conscious, effortful processes (e.g., "ballparking" before calculating). These behaviors, again, have been shown to be rare [11–13]. The other route is automatic, where something about the problem itself or the answer candidate arrived at from procedures students, correctly or not, believe are appropriate, captures their attention. This automatic capture of attention (or suspicion) we term "passive threshold". We propose that individual differences in conceptual knowledge (i.e. *numeracy* skills [14]) and concrete versions of problems–ones referencing real world situations rather than mere symbols and numbers–will increase the likelihood that people's passive threshold is triggered.

To test for factors that influence how people's passive thresholds are triggered, we asked participants to complete math problems using an onscreen calculator that was programmed to produce errors, or "lies", on certain problems. Previous studies have successfully used this manipulation to determine the extent to which highly numerate individuals defer to calculators [15–16]. Here, the "lying" calculator allowed us to create scenarios in the laboratory that mirror real world contexts where calculators give incorrect answers due to typos, employment of incorrect procedures, or from not knowing how some mathematical concepts are handled by the calculator (e.g., order of operations). We measured whether participants' passive thresholds were trigged by the "lying" calculator in three ways. First, to measure whether students became suspicious at any point during the entire study, we measured whether they explicitly reported suspicion when given the chance to do so at the end of the study. Second, on a trial-by-trial basis, we measured whether a participant corrected (i.e., overrode) the calculator's "lies", and whether a participant re-checked their keystrokes after the calculator "lied" to them.

We predicted their passive threshold would be triggered, as indicated by these suspicion measures, more frequently when: (a) concrete problems are used instead of abstract ones, due to increased activation of relevant knowledge, (b) when calculator answers are shown after a delay, hence encouraging students to potentially attempt mental calculation before they see the answer, and (c) when a participant has relatively greater knowledge of mathematics (i.e., higher numeracy scores).

By presenting math problems in concrete formats, it may help people activate the correct concepts based on their knowledge of how the concrete domain works. Such manipulations have been shown to improve performance both in logical reasoning tasks [17–18] and mathematics [19–24]. For temporal delay, we simply manipulated how long the calculator took to display answers. Introducing temporal delays in calculator output presumably affects the time and effort trade-offs students make when deciding to use a calculator rather than compute something mentally. Participants might become impatient and begin considering computing the problem mentally or perform a partial, "ballpark" calculation. Both scenarios should lead to more suspicion behaviors when "lies" are presented. These hypotheses are consistent with recent studies finding that participants tend to use mental computation and memory-based strategies when faced with lengthier delays (e.g., 4s [25], 2s [26]).

Finally, in addition to our experimental manipulations, we tested how "lie" detection is related to individual differences in numeracy. Numeracy scales capture individual differences in participants' background knowledge of mathematics and how easily they deploy such knowledge [27]. Thus, in the present study, it provides a critical measure of whether individuals are able to deploy mathematical concepts accurately, which would be a prerequisite for them being able to detect calculator errors. Numeracy is more than a simple prerequisite, however. Since the scales explicitly measures the ease with which these concepts are deployed, individual differences in numeracy should also be directly related to participants' passive thresholds for detecting errors. To measure numeracy, we used the popular *Expanded Numeracy Scale* [14]. To provide additional measurement on a broader range of concepts, we included questions created by Mulhern and Wylie [28]. These were originally used to diagnose mathematical competencies required among psychology undergraduates to begin training in quantitative methods courses. They are also more difficult for college students than those on the *expanded numeracy scale*, providing greater discriminability of skills than the *expanded numeracy scale* alone would allow.

## Study 1

The goal of Study 1 was to test how presentation (concrete vs abstract), delay, and individual differences in numeracy trigger students' passive thresholds, as indicated by our measures of suspicion behavior. To this end, we had participants complete a series of math problems in which an onscreen calculator was programmed to "lie" by 15% in two of the *variable presentation* (VP) questions (described in detail below). In the delay conditions, the calculator displayed answers to the VP questions after a 4 second delay. In the concrete conditions, VP questions were presented as word problems, making reference to familiar scenarios. In the abstract conditions, these same problems were framed in terms of numbers, symbols, and minimal words. We predicted that, in conditions where the calculator "lies", the effects of concrete format, delayed calculator displays, and numeracy would increase the probability of suspicion behavior. Additional conditions where the calculator did not "lie" served as a baseline for comparison and were not included in analyses of suspicion behavior. We also examined whether numeracy predicted whether people would tend to opt-out of using their calculator in the first place.

## Method

### Participants

Two-hundred and forty undergraduate students (30 per condition: See below) from The University of Louisiana-Lafayette (ULL) were recruited to participate in exchange for course credit. They were randomly assigned to one of 8 conditions resulting from a 2 (calculator: lie,

no lie) x 2 (delay: delay, no delay) x 2 (presentation: abstract, concrete) factorial, between-subjects design. ULL's Institutional Review Board approved the study. Participants were given a paper informed consent form to read and sign before beginning the study.

## Measures

**Variable Presentation (VP) questions.**    The VP section consisted of six math problems presented in either abstract or parallel concrete formats (see S1 Table). In the normal, non-lying condition, the calculator always showed the correct answers. For the "lie" conditions, the calculator was programmed to "lie" on two of the six problems. The other problems were included so that participants could become acquainted with the calculator and not be "lied" to twice in a row. An example of the different presentation formats would be "15% of 21 = _____" compared to, "You have just finished eating dinner and the bill is $21. You want to leave a 15% tip. How much would the tip be?" In the "lying" calculator conditions, the calculator was programmed to always add 15% (rounded to the nearest integer) on the third and fifth problems. This rounding was included because, on the third problem, a number with fractions would not have resulted from computing the difference between two whole numbers. The concrete version of the third problem read, "Your grandmother was born in 1942, how old was she in 1994?" whereas the abstract version read, "What is 1,994 minus 1,942?" Given the correct input, a normal calculator displayed the correct answer of '52' whereas the "lying" calculator displayed the incorrect answer of '60'. The concrete version of the fifth problem read, "Shelby is a ULL student who just paid $6,766 in tuition for 4 classes (each 3 credit hours). How much did she pay for each credit hour?" whereas the abstract version read "6,766 divided by 12 = _____." Given the correct input, a normal calculator displayed the correct answer of '563.8$\underline{3}$' whereas the "lying" calculator displayed the incorrect answer of '648'.

**Expanded numeracy scale.**    The expanded numeracy scale, developed by Lipkus and colleagues [14], is an 8-item test used by many researchers to quickly assess people's general conceptual understanding of risk-relevant, numerical information. The questions are mostly presented in terms of health-related scenarios.

**Mulhern and Wylie questions.**    To probe for broader conceptual knowledge, especially for learning statistics, and increase the range of difficulties of the problems overall, we added the questions created by Mulhern and Wylie [28] (henceforth, MW) to the study. These 20 questions included calculation of decimals and fractions in more depth than the *expanded numeracy scale*, as well as algebraic reasoning, graphical interpretation, proportionality, probability/sampling, and estimation. The two numeracy measures were summed and grand-mean centered for all analyses.

**Suspicion behavior.**    Suspicion behavior (or "lie" detection) was measured in three ways. Reporting suspicion towards the calculator at the end of the study was a direct measure, though it is one that does not capture problem-by-problem variations in suspicion. There were also two indirect measures. First, students "overrode" a calculator "lie" when they used the calculator on a target problem, typed out the entire problem, saw the erroneous answer, but entered the correct answer (and no other) as their answer to the problem. "Using" the calculator was defined as typing the full problem into the calculator and pressing "=" at the end. Merely clicking "clear", for example, did not count as using the calculator for the problem. Nearly all participants used the calculator for the first 2 VP problems and entered their calculator output as the answer. If a student does otherwise on the third VP item, one of the "lying" problems, they may not be confident enough in their doubts to override it, so we defined "rechecking" as typing out the operations for a problem in the calculator at least twice. The calculator would continue to produce the same error on "lie" problems, regardless of how many

times a participant entered the operations. Participants almost never "re-checked" outside the critical "lie" problems. The two indirect measures were combined in a disjunctive rule to code participants as being suspicious or not on individual problems. In other words, if participants re-checked or overrode on a problem, they were coded as being suspicious on that problem.

## Procedure

Participants were told that a tutoring program was being developed and they are helping with pilot testing. All questions were presented on a computer monitor. Answers were entered in a text field displayed beneath each problem or by selecting the correct answer on multiple choice problems. The three sections described above (VP questions, *expanded numeracy scale*, and MW questions) were presented in randomized order. Apart from the VP questions, which have a fixed order, questions within each section were also randomized. Participants were allowed to take as long as they desired to complete the study. Combinations of the delay and presentation factors applied to each VP problem. For instance, in the delay conditions, the calculator showed answers after a delay on each problem, not only on the critical problems.

An on-screen, four function calculator appeared during the VP section. In the "lie" conditions, it was programmed to always display an incorrect answer on the third and fifth problems. These were the critical problems, where accuracy and keystrokes were coded as indicating suspicion. The other problems in the VP section required more complicated procedures that would likely produce trial-and-error among a sizable proportion of participants. Such scenarios would have led to theoretically uninteresting cases of skepticism towards the calculator as "lies" grew exponentially with each calculation. We also wanted participants to be familiarized with the calculator before it "lied" to them.

When participants submitted their answers to the final problem, a page appeared, reading, "Thank you for participating in this study. Please provide any feedback regarding your experience in the box below." Afterwards, they were given a debriefing form, explaining the nature of the study and divulging the deception used therein.

A paper consent form was placed at each participant's station, which they were asked to read and sign before beginning the study. The researchers never gave instructions on whether to use the consent form as scratch paper, though some participants did so spontaneously. When asked, our protocol was to affirm that participants may use the paper or, if they complained there was no scratch paper, to suggest they use the consent form. Generally, we tried not to influence their behavior, preferring to observe rather than manipulate their natural strategy preferences. Even if a few participants surreptitiously used their own calculator (on their smartphone), we could tell from the absence of keystrokes when analyzing their data. All analyses consequently take into account whether students used their onscreen calculator.

## Results

We assessed the reliability of the combined numeracy scores with the Kuder-Richardson (KR) equation, which resulted in a moderately high 0.72. Only 1.67% of the participants in the non-"lying" conditions rechecked their keystrokes on VP3 and 97.5% answered correctly. For VP5, no one rechecked their keystrokes and 86.67% answered correctly. These results indicate that VP3 and VP5 are not inherently difficult to solve via calculator and that participants do not have an appreciable baseline level of re-checking their keystrokes that would confound the results from the "lying" conditions. Further on, we show that the presence of a delay likely did not cause students to opt-out of using their calculator, including those in the non-"lying" conditions. In the "lying" conditions, suspicion behavior was surprisingly rare. Only 20 out of 94

(21.28%) of calculator users showed suspicion on VP3 while 8 out of 109 (7.34%) users did so on VP5.

A multilevel logistic model was used with suspicion as the dependent variable and participants as a random effect, controlling for which problem they were answering. Suspicion was coded as 1 if one or both problem-level suspicion behaviors were exhibited and as 0 if neither were. Only participants that used the calculator on both problems were included in the analysis. We therefore excluded 28 of the 120 participants (23.33%, 12 from delay conditions, 16 from non-delay conditions), leaving 92 overall.

Neither the coefficient for presentation ($\gamma$ = 0.34, *SE* = .66, *p* = .608) nor delay ($\gamma$ = -0.46, *SE* = .66, *p* = .485) were significant. Participants were significantly less likely to show suspicion behavior on VP5 compared to VP3 ($\gamma$ = -1.82, *SE* = .35, *p* < .001), reflecting its greater difficulty. Higher numeracy increased the likelihood of suspicion behavior ($\gamma$ = 0.20, *SE* = .08, *p* = .011).

Four of the 120 (3.33%) participants explicitly reported suspicion at the end of the study. All of them were in abstract presentation conditions. Half were in delay conditions. Because so few people reported suspicion at the end of this study, no statistical analyses were conducted.

## Numeracy comparisons between users and non-users

To determine if numeracy influenced whether students opted-out of using their calculator, we ran a multilevel logistic regression model predicting calculator use with participants as a random effect and each VP problem nested within participants. The coefficient for numeracy was not significant ($\gamma$ = 0.04, *SE* = .06, *p* = .190), suggesting that numeracy did not influence the overall likelihood of using the calculator in the first place. Only 1 participant opted-out of using their calculator on all 6 problems. This participant was in a "lie", abstract, no-delay condition.

The presence of a delay in some conditions may have played a role in whether people used their calculator on a given trial. Students might show more willingness to engage in effortful processing if the calculator was less convenient to use. For instance, they may compute more problems mentally or "ballpark" what the answer should be. Our primary analysis examined whether the delay manipulation would influence suspicion behavior. To examine more directly whether it influenced calculator use in general, we conducted a multilevel logistic regression model with participants as a random effect, with calculator use on a given problem as the dependent variable, and numeracy scores, calculator condition, delay condition, presentation condition, and which problem they were on for a given trial as independent variables. The coefficient for delay was not significant ($\gamma$ = 0.13, *SE* = .29, *p* = .661). This result suggests that the delay manipulation probably did not cause students to use their calculators less frequently.

## Discussion

Our predictions about presentation effects were not supported in this study. The delay manipulation also failed to induce students into becoming suspicious at higher rates. Our hypotheses that greater numeracy predicted greater probabilities of exhibiting suspicion behavior were supported, however.

The absence of a delay effect may diverge from other studies examining the effect of delay on calculator use. For example, Walsh and Anderson [25] found that, under some conditions, participants' choice to use a calculator is influenced by the presence or absence of a delay. Their participants were explicitly paid as a function of how much time they took on each problem, however. We also used problems with a larger variety of difficulties whereas Walsh and

Anderson used a few variants of the same problem type: multiplying multi-digit numbers. Pyke and LeFevre's [26] study investigated people's ability to learn alphabet arithmetic (e.g., A + 2 = C) depending on whether they were computing the answer, retrieving it, or attempting to retrieve it before using a calculator. In this latter case, participants were shown a problem (e.g., B + 6 = ?) and were given 2 seconds (too little time to compute) to attempt retrieving the answer before they were required to use a calculator that appeared when time was up. Unlike both of these studies, we left participants with no time pressure and thus the effect of delay may not have influenced behavior. A future study could return to the delay manipulation, but in our context, where the emphasis was on examining how external factors trigger people's passive threshold, it was less relevant to do so.

What was most surprising about these results was the rarity of suspicion behavior. Only 20% and 7% of participants, on VP3 and VP5 respectively, showed suspicion behavior. It may be that the "lies" shown in this study were too subtle for most people to notice. For the next study, we increased the amount of error added to "lie" problems and the number of problems displaying "lies" overall.

## Study 2

For Study 2, we wanted to increase the frequency of suspicion behavior above the small amount observed in Study 1. We examined higher "lie" levels over a number of pilot studies (total $n = 61$; 30%, 60%, 90%, and 120% lie levels were used). One-hundred twenty percent was the only level that substantially increased suspicion behaviors. This would also make the concrete version of VP3 patently absurd. It reads, "If your grandmother was born in 1945, how old was she in 1994?" The difference between two dates within the same century cannot exceed 100 years and most humans do not live to be 114 years old regardless.

For the next two studies, we also added problems where the calculator "lied" categorically instead of by a magnitude. For example, the calculator showed answers that were negative that should have been positive. Magnitude "lies" have some minimal, expected characteristics. For instance, the "lying" calculator on VP5 will say that '6,766 / 12' equals a positive number that is lower than 6,766 and higher than 12. The number shown is still wrong, but adheres to some minimal, logical rules. Violating such rules may produce greater suspicion behavior, revealing a boundary condition on our initial findings. On the other hand, if the difference between the two types of "lies" do not produce a difference in lie detection rates, we will have found our results are robust across more situations than our initial study had shown.

Finally, an additional problem was included where the calculator did not lie but probed for whether students depend on the calculator to apply the negative of a squared expression or do so mentally.

## Method

In Study 2, participants were randomly assigned to view abstract or concrete versions of the VP problems. The calculator "lied" by 120% for both groups. The delay manipulation was dropped to increase the power of the study and because it had less theoretical concern compared to numeracy and concreteness.

### Participants

One-hundred and twenty undergraduate students from Texas Tech University (TTU) were recruited to participate in exchange for course credit. Half were randomly assigned to view the abstract version of the VP problems. The other half viewed the concrete versions. Unless otherwise noted, all other aspects of the procedure were the same. TTU's Human Research

Protection Program approved the study. Participants were given an informed consent form to read and sign before beginning the study.

## Measures

The same measures from Study 1 were used, but the VP section had the following problems added at the end.

**Conceptual problem #1 (CON1).** For CON1, students were asked to multiply '-9 x -7' and the calculator was programmed to always display the opposite sign of whichever answer the calculator would have normally shown.

**Conceptual problem #2 (CON2).** For CON2, participants were asked to multiply '0.5 x 0.9' and the calculator was programmed to always add 3 to the answer, resulting in '3.45', if the correct keystrokes were used. If participants knew that multiplying a number by 0.15 gave 15% of that number on VP1, and the keystrokes for many of the participants across these three studies suggests they do, then they should have been aware that CON2's answer cannot exceed 1. The problem is essentially asking what 50% of 90% is in decimal form, so it should never exceed 1 (i.e., 100%). Some students may solve this problem mentally, but for those who use the calculator, their answers and keystrokes will show whether they realized the number should be less than 1 (e.g., "overriding", re-checking).

**Probe problem (PROBE).** For PROBE, participants were asked to simplify the expression '-5.25$^2$'. It is important to note that the negative sign and '5.25$^2$' are "separate", so to speak. Mathematically, the negative sign will take "the opposite" of the square of '5.25', leading to a negative expression as the answer. We predicted most students would treat the problem as if it were (-5.25)$^2$. We wanted to observe the proportion of students who would enter the problem into a calculator, trusting it, perhaps, to apply all the relevant rules correctly on its own. An example of this would be a student entering, '-5.25 x -5.25'. A student who understands that the negative sign applies to the entire expression '5.25$^2$' may simplify it first then apply the negative sign afterwards. Alternatively, some students may enter '5.25 x 5.25' then mistakenly apply the negative sign to the answer manually. The crucial question, then, is whether there is a sizable proportion of students who multiply the two negative expressions and accept the positive answer (who are likely relying on the calculator to apply the sign rule) versus the amount of students who multiply the positive expressions and enter a negative answer (people who are using the calculator to compute '5.25$^2$' but applying the correct sign rule themselves).

## Procedure

The procedure was identical to Study 1, except for the following: a different "lie" level was used, 120%, which replaced the original 15%. People were only assigned to either an abstract or concrete presentation format during the VP section. No delay condition was present. The conceptual problems described above were added to the end of the VP section.

## Results

### Magnitude analysis

The Kuder-Richardson (KR) reliability for the combined numeracy scales was, again, moderately high, 0.76. Unlike Study 1, there were more participants showing suspicion, with 53 out of 87 (60.92%) calculator users showing suspicion for VP3 and 16 out of 110 (14.55%) doing so for VP5. The same analytical framework from Study 1 was used. Only participants who used the calculator on both magnitude "lie" problems were included in the magnitude analysis. This

excluded 39 out of 120 participants (32.5%, 20 from the abstract condition, 19 from the concrete condition), leaving 81 overall.

Being in the concrete condition ($\gamma$ = 2.12, $SE$ = .53, $p < .001$) and higher numeracy ($\gamma$ = 0.12, $SE$ = .06, $p$ = .032) both significantly increased the likelihood of exhibiting suspicion behavior. Consistent with Study 1, people were significantly less likely to exhibit suspicion behavior on VP5 compared to VP3 ($\gamma$ = -1.99, $SE$ = .51, $p < .001$), reflecting its greater difficulty.

## Conceptual analysis

Forty two out of 72 (58.33%) participants who used the calculator on CON1 showed suspicion while 26 out of 99 (26.26%) users did so on CON2. Like the magnitude analyses, only participants who used the calculator on both conceptual problems were included in the multilevel model. This led to excluding 53 out of 120 participants (44.17), leaving 67 for the multilevel model. Twenty four of the excluded participants were in the abstract condition during the magnitude section and 29 were in the concrete condition. The conceptual questions did not vary in presentation, however. For the conceptual analyses we added a predictor indicating whether they showed suspicion during the magnitude "lie" problems. Participants were less likely to show suspicion on CON2 compared to CON1 ($\gamma$ = -1.40, $SE$ = .40, $p < .001$), reflecting its greater difficulty. Higher numeracy increased the likelihood of showing suspicion ($\gamma$ = 0.12, $SE$ = .05, $p$ = .011). Prior suspicion had no effect ($\gamma$ = 0.59, $SE$ = .39, $p$ = .143).

To assess whether the difference between magnitude and conceptual lies played a role in suspicion behavior, we conducted a chi-square test on the frequencies of suspicion behaviors. We calculated frequencies for a two-by-two matrix with problem difficulty as columns and type (magnitude or conceptual problems) as rows. The test could not reject the null hypothesis of independence between the two factors.

Finally, 47 of the 120 (39.17%) participants explicitly reported suspicion at the end of the study. A logistic regression predicting suspicion reports at the end of the study showed no effect of condition ($b$ = 0.66, $SE$ = .40, $p$ = .103), but a significant effect of numeracy ($b$ = 0.12, $SE$ = .05, $p$ = .007).

## Numeracy comparison between users and non-users

To determine if numeracy influenced whether students opted-out of using their calculator, we ran a multilevel logistic regression model predicting calculator use with participants as a random effect and each VP problem nested within participants. The fixed effect for numeracy was not significant ($\gamma$ = 0.29, $SE$ = .25, $p$ = .234), suggesting that numeracy did not influence the overall likelihood of using the calculator in the first place. Only 1 participant opted-out of using their calculator on all 6 problems. They were in the concrete condition.

## Probe question

**PROBE.** Very few participants (5.83%, 7 out of 120) answered PROBE correctly. This made a two-sample $t$-test inappropriate. The $z$-scores in numeracy for these 7 participants were all positive, ranging from 1.19 to 9.19 with a mean of 4.05. These results suggest, unsurprisingly, that the few people who answered PROBE correctly were higher in numeracy. Most participants (61.67%) entered the positive '5.25' expression multiplied by itself. Presumably, they planned on applying the sign rule themselves. These students, however, had a strong tendency (93.24% of people using this strategy) to incorrectly assume the negative sign in '-5.25²' would be distributed as '-5.25 x -5.25', so that the negatives would cancel out and result in a positive product. The next most common strategy (18.33%) was for participants to explicitly

distribute the negative sign as '-5.25 x -5.25', possibly assuming the calculator would apply the correct sign rule itself. None of these students answered correctly. The remaining 20% of students who used other strategies always answered incorrectly as well. Their strategies ranged from multiplying '5.25' by two to multiplying a positive and negative version together.

## Discussion

Study 2 reproduced the effect of numeracy on suspicion behavior from Study 1 and extended these results by including conceptual "lies". The results of Study 2 also supported our hypothesis that presenting problems concretely will increase suspicion behavior, which we were not able to show in Study 1, when suspicion behaviors were so rare. We did not find that conceptual problems produce more suspicion behavior than magnitude ones. Like Study 1, we found that numeracy did not affect whether people tended to use their calculator. Regarding the PROBE question, there were around 18% of students who appeared to rely on the calculator to apply the correct sign rules and about 6% who only used the calculator to compute the square of '5.25' then apply the correct rules mentally.

Despite these results, it is still possible that participants were not motivated enough to perform well on these problems for them to notice errors (i.e., for the problems to trigger their passive threshold). To test this possibility, we repeated Study 2 with an incentive for better performance.

## Study 3

For the third study, we told participants at the beginning that 12 of the top 15% of performers would be randomly awarded a $5 Amazon.com gift card. This was done to determine whether a lack of motivation explains the effects observed in the previous two studies.

### Method

Study 3 was just like Study 2 except participants were offered the chance to win a gift card if they performed well, as described above.

### Participants

One-hundred twenty-two undergraduate students from TTU were recruited to participate in exchange for course credit. Half were randomly assigned to view the abstract version of the VP problems. The other half viewed the concrete versions. Unless otherwise noted, all other aspects of the procedure were the same. TTU's Human Research Protection Program approved the study. Participants were given a paper informed consent form to read and sign before beginning the study.

### Measures and procedure

The same measures and procedure from Study 2 were used, except for the added incentive described above.

### Results

#### Magnitude analysis

As a manipulation check, we compared numeracy scores between participants in Study 3 ($M = 16.63$, $SD = 4.55$) to those in the previous two studies ($M = 14.44$, $SD = 4.51$). Participants with the incentive offer had significantly higher scores (Welch's $t(207.28) = -4.61$, $p < .001$,

Cohen's $d$ = 0.48). This result suggests the incentive was effective at motivating higher performance. The Kuder-Richardson (KR) reliability for the combined numeracy scales was, again, moderately high, 0.75. Suspicion rates were comparable to Study 2. Fifty three out of 96 (52.21%) calculator users showed suspicion on VP3, while 28 out of 94 (38.30%) did so on VP5.

Only participants who used the calculator on both magnitude lie problems were included in the multilevel model, excluding 32 out of 122 participants (26.23%, 15 in the abstract condition, 17 in the concrete condition), leaving 90 overall. The same analytic framework from Study 1 and 2 were used. Like in the previous two studies, being in the concrete condition increased the likelihood of exhibiting suspicion behavior ($\gamma$ = 4.69, $SE$ = .87, $p <$ .001), as did numeracy ($\gamma$ = 0.24, $SE$ = .09, $p$ = .032). Also like the previous two studies, people were less likely to exhibit suspicion behavior on VP5 ($\gamma$ = -4.35, $SE$ = 0.43, $p$ = .001), reflecting its greater difficulty.

### Conceptual analysis

Calculator users were less likely to show suspicion behavior on CON1 in this study (29 out of 75, 38.67%), while users were more likely to show suspicion on CON2 (36 out of 94, 38.30%), compared to the previous study. The same approach from Study 2 was used to analyze suspicion behavior on the conceptual problems. Only participants who used the calculator on both conceptual problems were included in the analysis, excluding 57 out of 120 (47.5%) leaving 63 overall. Thirty-one of the excluded participants were in the abstract condition during the magnitude section and 28 were in the concrete version. The conceptual questions did not vary in presentation, however.

Like in Study 2, participants were less likely to show suspicion behavior on CON2 compared to CON1 ($\gamma$ = -21.69, $SE$ = .00, $p <$ .001), reflecting its greater difficulty. Numeracy scores were again associated with more suspicion behavior ($\gamma$ = 1.29, $SE$ = .30, $p <$ .001). Unlike Study 2, suspicion onset during the magnitude section was positively associated with suspicion during the conceptual section ($\gamma$ = 11.06, $SE$ = 2.81, $p <$ .001).

Like in Study 2, we conducted a chi-square test on the frequencies of suspicion behaviors in a two-by-two matrix with problem difficulty as columns and type (magnitude or conceptual) as rows. The test was significant this time ($\chi^2(1)$ = 5.53, $p$ = .019). Examining the standardized residuals, however, revealed that neither difficulty nor the type of "lie" contributed systematically in rejecting the null hypothesis of independence. Suspicion on VP3 and CON2 was slightly overrepresented, given the assumption of the null hypothesis, whereas suspicion on VP5 and CON1 was slightly underrepresented, assuming the null.

Finally, 54 of the 122 (44.26%) the participants explicitly reported suspicion at the end of the study. A logistic regression predicting suspicion reports at the end of the study showed a significant effect of condition ($b$ = 1.47, $SE$ = .42, $p <$ .001), where people were more suspicious for concrete problems, and numeracy ($b$ = 0.17, $SE$ = .05, $p <$ .001).

### Numeracy comparisons between users and non-users

To determine if numeracy influenced whether students opted-out of using their calculator, we ran a multilevel logistic regression model predicting calculator use with participants as a random effect and each VP problem nested within participants. The coefficient for numeracy was not significant ($\gamma$ = 0.09, $SE$ = .06, $p$ = .135), suggesting that numeracy did not influence the overall likelihood of using the calculator in the first place. Only 1 participant opted-out of using their calculator on all 6 problems.

*PROBE*. Again, very few participants (5.83%, 9 out of 122) answered PROBE correctly, which made a two-sample *t*-test inappropriate. The *z*-scores in numeracy for these participants ranged from -4.75 to 11.25 with a mean of 2.25. Three of the *z*-scores were negative while 6 were positive. Only one had an absolute value lower than two. These results suggest that, for the most part, participants with higher numeracy were able to use the correct rules, though some may have made lucky guesses about these rules. Another possibility is that the results from Study 2 were a Type 1 error and PROBE does not function well at discriminating people at different numeracy levels because of its extreme difficulty.

Again, most participants (71.31%) entered positive '5.25' multiplied by itself. Presumably, they planned on applying sign rules themselves. These students, however, had a strong tendency (89.66% of people using this strategy) to incorrectly assume the negative sign in '-5.25$^2$' would be distributed as '-5.25 x -5.25', so that the negatives would cancel out and result in a positive product. The next most common strategy (13.11%) was for participants to explicitly distribute the negative sign as '-5.25 x -5.25', possibly assuming the calculator would apply the correct sign rule itself. None of these students, or the remaining 15% of students who used other strategies, answered correctly.

## Discussion

Study 3 reproduced the results from Study 2, suggesting that the effects observed throughout these experiments are not due to a lack of motivation from participants. Concrete problem framing during the magnitude section increased the likelihood of suspicion, as did numeracy. Like Studies 1 and 2, we found no evidence that numeracy affects whether students will opt-out of using their calculator. Finally, we reproduced the results for the PROBE question, where more students appeared to rely on the calculator to apply the correct sign rules compared to the number that used the calculator only to compute the square of '5.25' then apply the correct rules mentally.

## Cross-study analyses

Although our primary research questions are within-study questions regarding the effect of problem presentation, delay, and numeracy, there are a number of differences across studies (e.g., lie magnitude) that could potentially provide additional information about factors that influence critical calculator usage. For this reason, we conducted a set of additional cross-study analyses to examine (a) the effect of lie magnitude and incentive levels, (b) whether, across studies, suspicion behavior affected calculator usage, and (c) whether differences in speed of answering questions (a potential measure of mathematical fluency) affected calculator usage.

To examine how suspicion behavior is affected by the different lie and incentive levels, suspicion behavior was regressed on problem presentation (abstract or concrete), problem (VP3 or VP5), study (1, 2, or 3), and participants' numeracy. Participants were included as a random effect and only those who were in a non-delay condition from Study 1 were included in the analysis.

Results revealed that suspicion behavior was significantly more likely to occur for people in Study 2 compared to the baseline of Study 1 ($\gamma$ = 1.42, *SE* = .60, *p* = .018). This same effect was observed for Study 3 compared to Study 1 ($\gamma$ = 1.98, *SE* = .59, *p* < .001). The difference in suspicion behavior between Study 2 and 3 was not significant ($\gamma$ = -0.56, *SE* = .44, *p* = .207). A second model included an additional study by condition interaction term to test whether the effect of condition (abstract of concrete) differed across studies. The coefficients for Study 2 ($\gamma$ = -0.35, *SE* = .79, *p* = .656) and 3 ($\gamma$ = 0.02, *SE* = .77, *p* = .978) both became insignificant while the Study 2 by condition ($\gamma$ = 3.54, *SE* = .1.15, *p* = .003) and Study 3 by condition ($\gamma$ = 3.88,

*SE* = 1.13, *p* < .001) interaction coefficients were both significant. These interaction coefficients were not significantly different from each other (γ = -0.35, *SE* = .88, *p* = .692). These results suggest that the 120% "lie" level from Study 2 and 3 alone did not significantly affect suspicion behavior. The addition of concrete problem presentations likely accounted for the additional suspicion behavior. The presence of an incentive in Study 3 also did not increase suspicion behavior compared to Study 2, even when accounting for potential differences in presentation effects between studies.

Participants who became suspicious on VP3 may have used their calculator less frequently on subsequent problems. The following analysis was conducted to determine whether students tended to stop using their calculator across the different lie magnitudes and incentive levels across the studies while controlling for variation among participants and problems. We conducted a multilevel logistic regression model predicting calculator use (binary; used or did not use) on the problems that came after VP3 (VP4, VP5, VP6) on all participants. We included covariates for the problem they were on, the study they were in (1, 2, or 3), and two binary indicators for whether they became suspicious on VP3 or VP5. Subjects were included as a random effect. Neither being in Study 2 (γ = 0.15, *SE* = .19, *p* = .433) nor Study 3 (γ = -0.15, *SE* = .19, *p* = .423) was associated with decreases in calculator usage relative to Study 1. The effect of suspicion onset on calculator usage at VP3 was not significant (γ = 0.07, *SE* = .19, *p* = .704), and neither was the effect for VP5 (γ = 0.20, *SE* = .26, *p* = .426). These results suggest that suspicion did not affect calculator usage between studies despite suspicion behavior significantly increasing in Study 2 and 3 relative to Study 1.

Finally, we wanted to examine whether procedural fluency could predict calculator usage where conceptual skill could not. Numeracy scales tend to focus on assessing conceptual understanding rather than procedural fluency. These constructs are likely correlated but dissociable. Students can potentially score high on a numeracy scale while being slow with, or distrust, their mental calculations. Because the program used in these studies measured how long students took on each question, we were able to use the average time spent solving the *expanded numeracy scale* problems as a proxy for procedural fluency. This measure was not a significant predictor of using the calculator on a given problem for Study 1 (γ = 0.01, *SE* = .02, *p* = .713) or 2 (γ = -0.01, *SE* = .03, *p* = .829), but was for Study 3 (γ = 0.05, *SE* = .02, *p* = .011). So, participants that took longer to solve the *expanded numeracy scale* problems, and perhaps had poorer procedural fluency, were more likely to use their calculator on a given problem during Study 3, but not during the other studies. This could be an indication that the incentive caused participants to apply their procedural fluency to a greater degree. Some caution is in order though because the average time taken to solve the *expanded numeracy scale* questions is not a recognized, standardized measure of procedural fluency. A new study focusing exclusively on this question would provide more definitive answers.

## General discussion

The present study demonstrated that a non-trivial proportion of university students do not know the correct procedures needed to answer common math problems and are often unaware of this fact. This was evident, for example, from the observation that roughly 15% of students throughout these studies did not know offhand how to calculate 15% of 21. We used a "lying calculator" to create situations that should trigger students' passive threshold. In other words, it should have captured their attention and create skepticism. Our results suggested that higher numeracy skills increase the likelihood that students' attention will be captured by nonsensical results, such as a candidate answer suggesting a woman born in 1942 was 114 years old during 1994.

We also found that framing problems in a concrete format increased the likelihood of triggering students' passive threshold. Instructors often frame new concepts in concrete situations that students have some familiarity with, which may be especially helpful, and sometimes necessary, for students with lower numeracy skills. It should be noted, however, that we are not advocating the sole use of concrete framing. On the contrary, the goal of teaching mathematics is to develop abstract, domain-general concepts. It has also been shown that it is probably best to start with problems based in concrete applications *but to transition ultimately to abstract problems* [29].

Throughout these studies, we observed no evidence that numeracy influences whether students opt-out of using their calculator, even for easy problems like '-9 x -7 = ___' and '1994–1942 = ___'. This could be interpreted as evidence that highly numerate students will tend to use a calculator, even when they do not need one. Future studies could also examine whether procedural fluency of mental computation predicts calculator usage to potentially add nuance to these results.

From Study 2 onward, we hypothesized that conceptual "lies" would produce more suspicion behavior than magnitude-wise errors. The results did not support this. Magnitude "lies" adhered to some basic, logical rules such as being a positive number and being within the correct order of magnitude. For instance, the difference between 1942 and 1994 was not in the thousands, millions, or hundred thousandths. Dividing 6,766 by 12 resulted in a number larger than 12 but smaller than 6,766. The conceptual "lies", on the other hand, were answers that disobeyed such rules. Since this made no difference in suspicion behavior, we concluded that the difficulty of solving the problem and its degree of concreteness are more influential factors.

A similar line of work has examined operation sense, a tendency to evaluate the results of mathematical operations in light of general principles such as the sum being larger than the numbers added to create the sum [30–32]. Past research has shown that people are sensitive to constraints on the result of an operation conditional on its inputs [33–34], e.g., a number multiplied by 5 should end in a 5 or a zero [35]. Yet, in Study 1, most participants did not show any suspicion over the proposition that '1994–1942' equals 60 instead of 52, despite the proposed answer ending in a 0 rather than an even number. Our results suggest that college students may either lack operation sense for some of the problems used here, deploy such knowledge to a lesser degree while using calculators, or a combination of the two. Future studies could use the "lying calculator" procedure to investigate these possibilities further.

This study also contributes to a growing literature examining how cognitive processes are influenced by technologies such as search engines [36–37], smartphones [38], and cameras [39]. The mere presence of someone's cell phone may even occupy people's attention enough to decrease working memory performance [40]. One suggestion [36] is that the presence of an external technology (e.g., a search engine) that gives people the ability to perform tasks they otherwise could not (e.g., search and retrieve information from the internet) gives them an inflated evaluation of their internal, mental abilities (e.g., general knowledge). Researchers in this area could adopt the procedures used here to investigate how mathematical cognition is affected by the ubiquity of calculators on smart phones.

The difficulty of the problems used in this study likely had a strong effect on whether students would show suspicion behavior, as is evident from differences between VP3 and VP5 as well as CON1 and CON2. This finding is consistent with another study that found objective difficulty increases cognitive offloading [41]. It is important to view the results from the present study in light of what students are capable of computing mentally and how easily they are able to do so. Future studies could focus more specifically on this relationship.

For this study, we assumed participants would experience meta-cognitive difficulties in monitoring their knowledge for routine problem solutions. The focus here was on how to

ameliorate this effect rather than explain its source. Future studies could investigate these origins using a "lying" calculator procedure like our own. Researchers could perform a cross-sectional study replicating a version of this study on adults of a broader age range to see whether suspicion behavior increases with age or self-reported habitual calculator usage. A longitudinal study could examine how cohorts of students potentially off-load mental computations onto calculators more over time. Previous research suggests a trend like this might be observed. An oft-cited meta-analysis [42], mostly including studies from the 1980s and 90s, failed to find negative effects of calculator use within the classroom on K-12 learning outcomes. More recent findings, however, suggest a negative association between calculator use and mathematical skill in K-12 curricula as well as habitual calculator use more broadly [43–46].

Finally, this study analyzed calculator usage, but the results may also apply to other domains, such as spreadsheet software, statistical packages, and decision aids.

## Limitations

The calculators in these studies only "lied" to participants up to four times, and only half of these had parallel abstract and concrete versions shown between-subjects. Future studies should use more varied stimuli to test the generality of the effects reported here. As noted before, it would be particularly useful to examine how problem difficulty affects meta-cognitive processes related to using the calculator.

Our protocol for instructing participants emphasized not interfering with their strategy choice. This decision gave us information about people who chose to use calculators, insights into their passive threshold, and the relationship between conceptual-mathematical knowledge and deployments of such knowledge, but at the expense of knowing less about the relatively few students who opt-out of using the calculator. We do not know how these students would have responded to its "lies". However, a decisive majority of students used the provided calculator on most problems. The non-users, therefore, make up a smaller proportion of the population of interest. Only three participants across these studies did not use the provided calculator at all. We also found no evidence of numeracy being positively associated with opting-out of calculator usage. It may still be valuable to know why certain students opted-out and how they would react to calculator lies. A future study could adapt the choice/no-choice method created by Siegler and Lemaire [47]. This would entail having participants go through a "free choice" condition, where they can use whichever strategy they like, followed by a number of "forced choice" conditions, isolating each available strategy for the task. In the present studies, we chose to observe rather than manipulate people's natural strategy selection, essentially with everyone in a "free choice" condition.

The combined numeracy scales showed reliabilities within ranges traditionally considered to be acceptable and in line with those observed by the original study validating the *expanded numeracy scale* by itself [14]. This is surprising, given that we added the Mulhern and Wylie questions to create a greater range of difficulty to the questions. This was likely to decrease the overall reliability of the scale. From a Classical Test Theory perspective, a more reliable measure is ideal. From an Item Response Theory perspective, it is desirable to include items with a greater range of difficulty to discriminate between people with minute differences in the latent trait (e.g., numeracy), possibly at the expense of reliability. So long as the scale is unidimensional or has a simple, well understood factor structure, we do not see an issue in making this trade-off.

## Conclusions

When students unknowingly use the incorrect procedures on a calculator, which is something we observed on many problems throughout these studies, there is a non-trivial number who

will not notice the implausibility of the answers displayed. Results from these studies suggest that more numerate students may be able to detect errors (e.g., incorrect operations, typos) when using a calculator. Apparently, these errors must be blatant for an appreciable number of students to notice them. If problems are presented concretely, where people's prior knowledge can come to bear on candidate answers, this will likely help them detect errors.

## Supporting information

**S1 Table. Abstract and concrete versions for the variable presentation problems.** (DOCX)

## Acknowledgments

Special thanks to Claude Cech for inspiring the initial study reported here.

## Author Contributions

**Conceptualization:** Mark LaCour, Norma G. Cantú, Tyler Davis.

**Data curation:** Mark LaCour, Norma G. Cantú.

**Formal analysis:** Mark LaCour.

**Investigation:** Mark LaCour.

**Methodology:** Mark LaCour.

**Project administration:** Mark LaCour, Norma G. Cantú.

**Resources:** Tyler Davis.

**Supervision:** Tyler Davis.

**Writing – original draft:** Mark LaCour, Tyler Davis.

**Writing – review & editing:** Tyler Davis.

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
