## [Decision Letter · Decision Letter 0]

7 Aug 2019

PONE-D-19-15769

When calculators lie: A demonstration of uncritical calculator usage among college students and factors that improve performance

PLOS ONE

Dear Mr. LaCour,

Thank you for submitting your manuscript to PLOS ONE. After careful consideration, we feel that it has merit but does not fully meet PLOS ONE’s publication criteria as it currently stands. Therefore, we invite you to submit a revised version of the manuscript that addresses the points raised during the review process.

Please find below the reviewer's comments.

We would appreciate receiving your revised manuscript by Sep 21 2019 11:59PM. To enhance the reproducibility of your results, we recommend that if applicable you deposit your laboratory protocols in protocols.io, where a protocol can be assigned its own identifier (DOI) such that it can be cited independently in the future. For instructions see: http://journals.plos.org/plosone/s/submission-guidelines#loc-laboratory-protocols

We look forward to receiving your revised manuscript.

Kind regards,

Valerio Capraro

Academic Editor

PLOS ONE

Journal Requirements:

Additional Editor Comments (if provided):

I have now collected one review from one expert in the field. Unfortunately, I was unable to find a second reviewer. However, the one review I was able to collect is very detailed and, therefore, after reading the manuscript, I have decided to make a decision with only this review. As you will see, the review is very positive, but suggests a number of revisions before publication. Therefore, I would like to invite you to revise your paper following the reviewer's comments. I am looking forward for the revision.

Reviewers' comments:

Reviewer's Responses to Questions

**Comments to the Author**

1. Is the manuscript technically sound, and do the data support the conclusions?

Reviewer #1: Yes

2. Has the statistical analysis been performed appropriately and rigorously? 

Reviewer #1: Yes

3. Have the authors made all data underlying the findings in their manuscript fully available?

Reviewer #1: Yes

4. Is the manuscript presented in an intelligible fashion and written in standard English?

Reviewer #1: Yes

5. Review Comments to the Author

Reviewer #1: Review: When Calculators Lie 2019 (Plos1)

Summary: This interesting research explored factors that influence whether or not solvers detect when calculators yield incorrect/implausible answers. This can happen ‘in the wild’ when typos are made and/or incorrect procedures are used, and, more theoretically, solver’s sensitivity to such errors arguably reflects math aptitude and understanding. The authors hypothesized (correctly) that an individual’s numeracy (conceptual understanding of math) would predict their sensitivity to such errors. Further, presenting problems in concrete contexts (word problems) vs. abstract form increases sensitivity at least in the case of large errors (Study 2,3). In Study1, the delay of calculator answer production -- to allow users to form expectations about the answer -- did not seem help them notice incorrect answers. [Some further discussion/interpretation desired]. Nor did adding a $5 financial performance incentive seem to increase sensitivity, although this comparison was not (yet) tested statistically (Study 2 vs. Study 3). Error magnitude (15% off vs. 120% off) and type (magnitude error vs. conceptual error: e.g., product of two negatives being negative) were also considered.

This is interesting, novel research with theoretical and pedagogical relevance. It is connected to various existing lines of research, and interesting and motivated factors are considered. It is also generally well written.

Study 1: 8 groups: 2(calc: 15% lie vs. no-lie) x 2(delay) X 2 (presentation: concrete vs. abstract)

Within: lie probs vs. non-lie probs?

Study 2: 2 groups: 2(Presentation: abstract vs. concrete), 120% lying on magnitude lie problems

Within: Magnitude lies problems vs. Concept lie Problems?

There we no non-lie concept problems (baseline)

Study 3: 2 groups: 2(Presentation: abstract vs. concrete), 120% lying + $5 performance incentive

Within: Magnitude lies problems vs. Concept lie Problems?

1. Some key additional analyses are requested since some very interesting manipulations are across vs. within experiments. It seems feasible and would add considerable value to add statistical analyses combining/comparing experiments to better understand the influence of these interesting manipulations, for example:

a) magnitude of calculator error/lie (0%, 15%, 120%)

0% in no-lie group in Study 1, 15% in lie group in Study 1, vs. 120% in Study 2/3

just include the no-delay subset in Study 1 since there was no delay in Study 2/3

i) does lie magnitude affect Suspicion Likelihood

ii) does lie magnitude affect Calculator Use (e.g., less likely to use Calculator if lies big-time)

(on Q4-Q6 for those who used the calculator on Q3)

b) Performance Incentive: Study 2=no, Study 3=yes

i) does incentive affect Suspicion Likelihood

ii) does incentive affect Calculator Use (e.g., more likely to use calculator if money on the line?)

Study 2&3 could maybe even be described as a single experiment.

2. The use of the On-screen calculator use was *NOT* mandated/required for all VP questions. There are pros and cons to this design.

a) On the con side, allowing solvers to opt out seems can cause missing data if they opted not to use calculator on critical “Lie” questions in the lie condition? The people who didn’t use the calculator on those questions couldn’t be included in the analysis (since they never saw the error to be suspicious of). The people good enough at math to opt out of calculator use (and thus be excluded from the analysis) however, arguably includes a subset of the sample especially likely to be skilled/savvy enough to be suspicious of dodgy answers. This possibility/limitation should be more explicitly spelled out in the discussion about the rarity of suspicion frequency (overriding/rechecking). E.g., line 550

b) HOWEVER, on the pro side, calculator use could itself serve as an additional potential index of suspicion. Opting to decrease calculator use after exposure to errors is possible evidence of suspicion. Was an original intention of the design (allowing them the flexibility to opt out of calculator use) to possibly see if they would opt to use the calculator less on questions 4-6 (especially quest 6 after 2 lies) in the lie condition than no lie condition as another possible indicator of suspicion? Do the data cooperate? (Please do and report requested analysis in Point 1)a)ii) above).

3. Delay Factor (Study 1):

a) In Walsh & Anderson, if I recall correctly, calculator efficiency influenced the likelihood of them opting to use it (e.g., on subsequent trials) – so just as you tested to see if numeracy influenced peoples’ tendency to opt out of calculator use, also please test whether this delay factor influenced calculator use.

b) What proportion of people were excluded from suspicion analysis (as non-users) in delay group vs. no-delay group? If delay caused more people to be excluded, it is possible that the people likely to opt out of calculator use (in response to delay) may be the more skilled ones and thus the ones who would have been more likely to detect errors. This should be discussed as a possible reason for failure to find an effect of delay on suspicion (if it induced more calculator opt out ).

c) Additionally, in discussion section (Study 1 and/or general), could include other possibility as to why this hypothesized factor may have not had an effect (on suspicion) in this study, but did have influences (on other measures) in Walsh&Anderson (2009) and Pyke&LeFevre (2011). I might speculate that in Pyke&LeFevre (2011) the delay was before they could use the calculator, so they had nothing to do except try to mentally come up with the answer. However in the current research, I believe the delay came after they had already put in the effort of typing the problem into the calculator, so in that case they might not be inclined to further exert themselves to also then calculate mentally – rather they might just wait for the fruits of the labor (typing) they’d already invested.

4. Lie vs. Baseline Comparison (Study 1): -Line 244: Significance of Results are reported for various factors/coefficients (e.g., presentation format, delay, question number) – but not for the calculator (lie, no-lie) factor. Assume it is ns? Please report. Maybe the baseline condition wasn’t included in this analysis? However, since it was a manipulation there should be some statistical analysis comparing lie vs. no lie conditions.

5. End of Study Suspicion Reports: this info is reported for study 2 & 3, but not study 1

6. Numeracy and Calculator Use: Gen Discussion should consider why numeracy (as measured here) might not have influenced calculator use. It may have to do with the extent the measure focuses on conceptual knowledge vs. procedural fluency. Which is fine, as conceptual knowledge is likely more important/predictive for suspicion. Were the numeracy evaluations timed or time-limited? Measures related to time/speed related like calculation or retrieval “fluency” might better predict reliance on external computation devices. If there is timing (speed) data on for the numeracy tasks – this might predict calculator use. Another issue is that some people may have been using external calculators/smart phones. This should be controlled/prohibited in future research. Otherwise there is ambiguity in data interpretation – no keystrokes on on-screen calculator could mean they opted to mentally calculate OR use an external calculator.

7. Conceptual errors (Study 2): such conceptual errors and (detecting) sensitivity to them relates to some existing lines of research on “Operation Sense” by researchers such as Prather and (Dixon, Deets & Bangert). Some of that research should at least be mentioned/cited in the context of background and/or discussion

Minor Points:

-Abstract: after i.e., it should be “from” not “form”

-Line 123: Sentence structure is a bit long/awkward. Maybe break into two sentences: However, **numeracy is** more than a simple prerequisite. Since numeracy explicitly….

-Line 131: missing words “than the”: greater discriminability of skills **than the** expanded….

-Line 157/158: Since it is between groups design, recommend clarifying that in the no lie condition the calculator will tell the truth on all problems vs. in the lie condition the calculator will lie on two problems. (If my understanding is correct).

- Why round the answer to the nearest integer in the lie condition?

- Please clarify in procedure: If participants rechecked an answer (re-typed question into calculator) – would it provide correct answer this time or still give the same error?

-Line 241: if only participants that used calculator on both critical problems were included in analysis – please explicitly state what proportion of participants in the lie condition were omitted/included? Also do this for Study 2 and 3 please.

-Line 242/243: Why were 94 participants involved in suspicion percentages for VP3 and 109 participants for VP5? Is this now including participants who used calculator for each question but not nec. Both questions.

6. PLOS authors have the option to publish the peer review history of their article (what does this mean?). If published, this will include your full peer review and any attached files.

Reviewer #1: No

---

## [Author Response · Author response to Decision Letter 0]

7 Sep 2019

September 7, 2019

Dear Professor Valerio Capraro:

We are pleased to resubmit our manuscript, “When calculators lie: A demonstration of uncritical calculator usage among college students and factors that improve performance,” for consideration in PloS One. We thank the current and previous reviewers for their thoughtful observations. 

The review of our previous submission was largely positive, answering that we met all of PloS One’s primary review criteria. However, the reviewer also had a number of suggestions for additional analyses and clarifications that we believe have substantially improved the manuscript. We now include extensive discussions of how our delay manipulation departs from previous studies, three new cross-study analyses, and several clarifications and additional data reports. We hope that you now agree that the manuscript is now acceptable for publication at PloS One

We now address each of the reviewer’s comments point-by-point below.

1. Some key additional analyses are requested since some very interesting manipulations are across vs. within experiments. It seems feasible and would add considerable value to add statistical analyses combining/comparing experiments to better understand the influence of these interesting manipulations, for example: a) magnitude of calculator error/lie (0%, 15%, 120%) 0% in no-lie group in Study 1, 15% in lie group in Study 1, vs. 120% in Study 2/3 just include the no-delay subset in Study 1 since there was no delay in Study 2/3 i) does lie magnitude affect Suspicion Likelihood

We now include a Cross-Study Analyses section just before the General Discussion. We directly tested the effects of lie level and incentive on suspicion behavior across the studies as the reviewer suggested. Also at the reviewer’s suggestion, we excluded participants from the delay conditions of Study 1 to eliminate theoretically uninteresting variance between-studies. Because our measure of lie detection was only defined for trials where participants were lied to, we did not include participants in the “non-lying” conditions for Study 1.We have revised our methods and results sections throughout the manuscript to make this clearer. In regard to the suggested analyses, the second paragraph of the new Cross-Study Analyses section reads,

“To examine how suspicion behavior is affected by the different lie and incentive levels, suspicion behavior was regressed on problem presentation (abstract or concrete), problem (VP3 or VP5), study (1, 2, or 3), and participants’ numeracy. Participants were included as a random effect and only those who were in a non-delay condition from Study 1 were included in the analysis.

Results revealed that suspicion behavior was significantly more likely to occur for people in Study 2 compared to the baseline of Study 1 (� = 1.42, SE = .60, p = .018). This same effect was observed for Study 3 compared to Study 1 (� = 1.98, SE = .59, p < .001). The difference in suspicion behavior between Study 2 and 3 was not significant (� = -0.56, SE = .44, p = .207). A second model included an additional study by condition interaction term to test whether the effect of condition (abstract of concrete) differed across studies. The coefficients for Study 2 (� = -0.35, SE = .79, p = .656) and 3 (� = 0.02, SE = .77, p = .978) both became insignificant while the Study 2 by condition (� = 3.54, SE = .1.15, p = .003) and Study 3 by condition (� = 3.88, SE = 1.13, p < .001) interaction coefficients were both significant. These interaction coefficients were not significantly different from each other (� = -0.35, SE = .88, p = .692). These results suggest that the 120% “lie” level from Study 2 and 3 alone did not significantly affect suspicion behavior. The addition of concrete problem presentations likely accounted for the additional suspicion behavior. The presence of an incentive in Study 3 also did not increase suspicion behavior compared to Study 2, even when accounting for potential differences in presentation effects between studies.” (p. 25-26)

The reviewer also asked whether lie magnitude and incentive affected calculator use: ii) does lie magnitude affect Calculator Use (e.g., less likely to use Calculator if lies big-time) (on Q4-Q6 for those who used the calculator on Q3). We have added this analysis in the Cross-Study Analyses section as well. It reads,

“Participants who became suspicious on VP3 may have used their calculator less frequently on subsequent problems. The following analysis was conducted to determine whether students tended to stop using their calculator across the different lie magnitudes and incentive levels across the studies while controlling for variation among participants and problems. We conducted a multilevel logistic regression model predicting calculator use (binary; used or did not use) on the problems that came after VP3 (VP4, VP5, VP6) on all participants. We included covariates for the problem they were on, the study they were in (1, 2, or 3), and two binary indicators for whether they became suspicious on VP3 or VP5. Subjects were included as a random effect. Neither being in Study 2 (� = 0.15, SE = .19, p = .433) nor Study 3 (� = -0.15, SE = .19, p = .423) was associated with decreases in calculator usage relative to Study 1. The effect of suspicion onset on calculator usage at VP3 was not significant (� = 0.07, SE = .19, p = .704), and neither was the effect for VP5 (� = 0.20, SE = .26, p = .426). These results suggest that suspicion did not affect calculator usage between studies despite suspicion behavior significantly increasing in Study 2 and 3 relative to Study 1.” (p. 26)

The reviewer suggested we describe Study 2 and 3 as a single experiment: Study 2&3 could maybe even be described as a single experiment. We recognize that this could potentially make the results clearer for readers and gave the suggestion much thought. We hope to keep the two studies separate for the sake of transparency and to avoid any biases that may arise due to the results of Study 2 being known prior to conducting Study 3. To give some background, Study 3 was conducted after receiving a previous revise and resubmit decision here at Plos One. Thus the results of Study 2 had already been determined prior to Study 3, which makes combining them potentially fall under “data peaking”. We believe it is stronger and more transparent to describe the replication and extension as a separate study, as it was originally conceived, than combine them post hoc into a single study.

The next section discussed our decision to allow participants to opt-in or out of using their calculators freely. 2. The use of the On-screen calculator use was *NOT* mandated/required for all VP questions. There are pros and cons to this design. a) On the con side, allowing solvers to opt out seems can cause missing data if they opted not to use calculator on critical “Lie” questions in the lie condition? The people who didn’t use the calculator on those questions couldn’t be included in the analysis (since they never saw the error to be suspicious of)… This possibility/limitation should be more explicitly spelled out in the discussion about the rarity of suspicion frequency (overriding/rechecking). E.g., line 550 

We agree this is a limitation of the study that readers should be aware of. We regret this was not given greater emphasis in the original manuscript. We have added cautionary remarks throughout the manuscript and the issue been given greater emphasis in the limitations section. We have also now included the exact number of participants excluded from the magnitude and conceptual analyses for each study to potentially drive this point further home for readers.

This section, on pages 30-31, read as follows.

“Our protocol for instructing participants emphasized not interfering with their strategy choice. This decision gave us information about people who chose to use calculators, insights into their passive threshold, and the relationship between conceptual-mathematical knowledge and deployments of such knowledge, but at the expense of knowing less about the relatively few students who opt-out of using the calculator. We do not know how these students would have responded to its “lies”. However, a decisive majority of students used the provided calculator on most problems. The non-users, therefore, make up a smaller proportion of the population of interest. Only three participants across these studies did not use the provided calculator at all. We also found no evidence of numeracy being positively associated with opting-out of calculator usage. It may still be valuable to know why certain students opted-out and how they would react to calculator lies. A future study could adapt the choice/no-choice method created by Siegler and Lemaire [47]. This would entail having participants go through a “free choice” condition, where they can use whichever strategy they like, followed by a number of “forced choice” conditions, isolating each available strategy for the task. In the present studies, we chose to observe rather than manipulate people’s natural strategy selection, essentially with everyone in a “free choice” condition.”

The reviewer also said, The people good enough at math to opt out of calculator use (and thus be excluded from the analysis) however, arguably includes a subset of the sample especially likely to be skilled/savvy enough to be suspicious of dodgy answers. Our original analyses suggested there was no difference in numeracy between those who opted-in or out of using their calculator. We did, however, find some potentially interesting differences in calculator usage depending on the speed with which people completed the expanded numeracy section, which we describe in section five below.

b) HOWEVER, on the pro side, calculator use could itself serve as an additional potential index of suspicion. Opting to decrease calculator use after exposure to errors is possible evidence of suspicion. Was an original intention of the design (allowing them the flexibility to opt out of calculator use) to possibly see if they would opt to use the calculator less on questions 4-6 (especially quest 6 after 2 lies) in the lie condition than no lie condition as another possible indicator of suspicion? Do the data cooperate? (Please do and report requested analysis in Point 1)a)ii) above). 

We have conducted this analysis, and now report it in the manuscript. It seems that suspicion onset does not impact future calculator usage.

“Participants who became suspicious on VP3 may have used their calculator less frequently on subsequent problems. The following analysis was conducted to determine whether students tended to stop using their calculator across the different lie magnitudes and incentive levels across the studies while controlling for variation among participants and problems. We conducted a multilevel logistic regression model predicting calculator use (binary; used or did not use) on the problems that came after VP3 (VP4, VP5, VP6) on all participants. We included covariates for the problem they were on, the study they were in (1, 2, or 3), and two binary indicators for whether they became suspicious on VP3 or VP5. Subjects were included as a random effect. Neither being in Study 2 (� = 0.15, SE = .19, p = .433) nor Study 3 (� = -0.15, SE = .19, p = .423) was associated with decreases in calculator usage relative to Study 1. The effect of suspicion onset on calculator usage at VP3 was not significant (� = 0.07, SE = .19, p = .704), and neither was the effect for VP5 (� = 0.20, SE = .26, p = .426). These results suggest that suspicion did not affect calculator usage between studies despite suspicion behavior significantly increasing in Study 2 and 3 relative to Study 1.” (p. 26)

The next section concerned the delay effect from Study 1: 3. Delay Factor (Study 1): a) In Walsh & Anderson, if I recall correctly, calculator efficiency influenced the likelihood of them opting to use it (e.g., on subsequent trials) – so just as you tested to see if numeracy influenced peoples’ tendency to opt out of calculator use, also please test whether this delay factor influenced calculator use… c) Additionally, in discussion section (Study 1 and/or general), could include other possibility as to why this hypothesized factor may have not had an effect (on suspicion) in this study, but did have influences (on other measures) in Walsh&Anderson (2009) and Pyke&LeFevre (2011). I might speculate that in Pyke&LeFevre (2011) the delay was before they could use the calculator, so they had nothing to do except try to mentally come up with the answer. However in the current research, I believe the delay came after they had already put in the effort of typing the problem into the calculator, so in that case they might not be inclined to further exert themselves to also then calculate mentally – rather they might just wait for the fruits of the labor (typing) they’d already invested.

We discuss differences between our study and the two discussed by the reviewer that turned out to be crucial. A new paragraph in the manuscript reads,

“The absence of a delay effect may diverge from other studies examining the effect of delay on calculator use. For example, Walsh and Anderson [25] found that, under some conditions, participants’ choice to use a calculator is influenced by the presence or absence of a delay. Their participants were explicitly paid as a function of how much time they took on each problem, however. We also used problems with a larger variety of difficulties whereas Walsh and Anderson used a few variants of the same problem type: multiplying multi-digit numbers. Pyke and LeFevre’s [26] study investigated people’s ability to learn alphabet arithmetic (e.g., A + 2 = C) depending on whether they were computing the answer, retrieving it, or attempting to retrieve it before using a calculator. In this latter case, participants were shown a problem (e.g., B + 6 = ? ) and were given 2 seconds (too little time to compute) to attempt retrieving the answer before they were required to use a calculator that appeared when time was up. Unlike both of these studies, we left participants with no time pressure and thus the effect of delay may not have influenced behavior. A future study could return to the delay manipulation, but in our context, where the emphasis was on examining how external factors trigger people’s passive threshold, it was less relevant to do so.” (p. 13)

Returning to the delay manipulation in the present context, where emphasis is placed on low-time pressure, and free strategy choice, could be fruitful. In a pilot study investigating how much of a time delay this may require, however, we found no difference among 42 students in their calculator usage between a 2-, 4-, and 7-second delay on each VP problem. This is despite including two easier problems in the beginning (‘7 x 6 = ___’ and ’63 – 8 = ___’) and despite several of the VP problems requiring multiple computations. These results suggest it could take an extraordinary amount of wait time for students to choose a more effortful strategy. Currently, we do not include this pilot study in the paper to avoid overwhelming the reader as we don’t return to the delay manipulation in the later studies. However, we could add these details if the reviewer or editor would find them helpful.

The reviewer also asked how many participants were excluded from the delay conditions: b) What proportion of people were excluded from suspicion analysis (as non-users) in delay group vs. no-delay group? If delay caused more people to be excluded, it is possible that the people likely to opt out of calculator use (in response to delay) may be the more skilled ones and thus the ones who would have been more likely to detect errors. This should be discussed as a possible reason for failure to find an effect of delay on suspicion (if it induced more calculator opt out ).

A total of 28 participants were dropped from the magnitude analysis that tested the effect of delay. Of these 28, 12 were in a delay condition. Of the 92 who remained in the analysis, 48 were in a delay condition. We now list the number of participants dropped from each condition throughout the manuscript. 

In the fourth section, the reviewer inquired about the “no-lie” condition: 4. Lie vs. Baseline Comparison (Study 1): -Line 244: Significance of Results are reported for various factors/coefficients (e.g., presentation format, delay, question number) – but not for the calculator (lie, no-lie) factor. Assume it is ns? Please report. Maybe the baseline condition wasn’t included in this analysis? However, since it was a manipulation there should be some statistical analysis comparing lie vs. no lie conditions.

We did not include participants in the “non-lying” conditions in our analyses for a number of reasons. First, we wanted to measure suspicion behavior toward lies, so we didn’t think it would be appropriate to include people in our analyses who weren’t lied to. Second, one of the suspicion behaviors, overriding, can only occur if a participant is lied to. The other suspicion behavior, re-checking is possible without the lie, but only 2 out of the 120 (1.67%) of the participants in the non-“lying” conditions rechecked their keystrokes on VP3 and none of them did so on VP5. We included the “no lie” conditions so that we can assess baseline levels of participant double-checking their keystrokes, which was very rare, without being lied to, and to assess the difficulty of the problems with a non-lying calculator. These data helped us place more confidence in our inferences. For instance, we can rule out the possibility that VP3 was too difficult to solve mentally or with a calculator or that participants did not, in general, know which keystrokes should be used to solve VP5. We have added remarks throughout the manuscript, especially in the Methods and Results sections of Study 1 to clarify these points.

5. End of Study Suspicion Reports: this info is reported for study 2 & 3, but not study 1. We thank the reviewer for noticing this oversight. It has been corrected in the new manuscript.

The sixth section discusses our observation that numeracy did not affect calculator use: 6. Numeracy and Calculator Use: Gen Discussion should consider why numeracy (as measured here) might not have influenced calculator use. It may have to do with the extent the measure focuses on conceptual knowledge vs. procedural fluency. Which is fine, as conceptual knowledge is likely more important/predictive for suspicion. Were the numeracy evaluations timed or time-limited? Measures related to time/speed related like calculation or retrieval “fluency” might better predict reliance on external computation devices. If there is timing (speed) data on for the numeracy tasks – this might predict calculator use. 

We have added the following section in the Cross-Study Analyses and hopefully addresses these questions:

“Finally, we wanted to examine whether procedural fluency could predict calculator usage where conceptual skill could not. Numeracy scales tend to focus on assessing conceptual understanding rather than procedural fluency. These constructs are likely correlated but dissociable. Students can potentially score high on a numeracy scale while being slow with, or distrust, their mental calculations. Because the program used in these studies measured how long students took on each question, we were able to use the average time spent solving the expanded numeracy scale problems as a proxy for procedural fluency. This measure was not a significant predictor of using the calculator on a given problem for Study 1 (� = 0.01, SE = .02, p = .713) or 2 (� = -0.01, SE = .03, p = .829), but was for Study 3 (� = 0.05, SE = .02, p = .011). So, participants that took longer to solve the expanded numeracy scale problems, and perhaps had poorer procedural fluency, were more likely to use their calculator on a given problem during Study 3, but not during the other studies. This could be an indication that the incentive caused participants to apply their procedural fluency to a greater degree. Some caution is in order though because the average time taken to solve the expanded numeracy scale questions is not a recognized, standardized measure of procedural fluency. A new study focusing exclusively on this question would provide more definitive answers.” (p. 26-27)

“Another issue is that some people may have been using external calculators/smart phones. This should be controlled/prohibited in future research. Otherwise there is ambiguity in data interpretation – no keystrokes on on-screen calculator could mean they opted to mentally calculate OR use an external calculator.”

We agree that students should be kept from using their own calculators. The research assistants and lead author on the study do not recall any participants using their smart phone during the study, though participants could have just been furtive enough to use their phones without our noticing. One thing we have noted in the manuscript is that only 1 participant per study did not use the on-screen calculator on a single problem, e.g., 

“Only 1 participant opted-out of using their calculator on all 6 problems. This participant was in a “lie”, abstract, no-delay condition.” (p. 12)

7. Conceptual errors (Study 2): such conceptual errors and (detecting) sensitivity to them relates to some existing lines of research on “Operation Sense” by researchers such as Prather and (Dixon, Deets & Bangert). Some of that research should at least be mentioned/cited in the context of background and/or discussion. We thank the reviewer for introducing us to this fascinating and very relevant line of research. After doing some reading, we have added the following paragraph in the General Discussion section.

“A similar line of work has examined operation sense, a tendency to evaluate the results of mathematical operations in light of general principles such as the sum being larger than the numbers added to create the sum [30-2]. Past research has shown that people are sensitive to constraints on the result of an operation conditional on its inputs [33-4], e.g., a number multiplied by 5 should end in a 5 or a zero [35]. Yet, in Study 1, most participants did not show any suspicion over the proposition that ’1994 – 1942’ equals 60 instead of 52, despite the proposed answer ending in a 0 rather than an even number. Our results suggest that college students may either lack operation sense for some of the problems used here, deploy such knowledge to a lesser degree while using calculators, or a combination of the two. Future studies could use the “lying calculator” procedure to investigate these possibilities further.” (p. 29)

Finally, we have hopefully addressed the following Minor Points to the reviewer and editor’s satisfaction:

-Abstract: after i.e., it should be “from” not “form”

We thank the reviewer for catching this oversight. It has been corrected.

-Line 123: Sentence structure is a bit long/awkward. Maybe break into two sentences: However, **numeracy is** more than a simple prerequisite. Since numeracy explicitly....

We agree this sentence was too long and have broken it up into separate sentences.

-Line 131: missing words “than the”: greater discriminability of skills **than the** expanded....

We thank the reviewer for catching this oversight. It has been corrected.

-Line 157/158: Since it is between groups design, recommend clarifying that in the no lie condition the calculator will tell the truth on all problems vs. in the lie condition the calculator will lie on two problems. (If my understanding is correct).

The Measures section has been revised to make these clarifications.

- Why round the answer to the nearest integer in the lie condition?

We have added the following sentence to hopefully clarify this: “This rounding was included because, on VP3, a number with fractions would not have resulted from computing the difference between two whole numbers.” 

- Please clarify in procedure: If participants rechecked an answer (re-typed question into calculator) – would it provide correct answer this time or still give the same error?

It has now been clarified in the first Procedure section, and in some others, that the “lying” calculator indeed kept displaying a false answer for any computation so long as the participant was on one of the problems it was programmed to lie on.

-Line 241: if only participants that used calculator on both critical problems were included in analysis – please explicitly state what proportion of participants in the lie condition were omitted/included? Also do this for Study 2 and 3 please. 

These numbers have been added to each results section, which we also hope will drive home to readers the limitation that participants were allowed to freely opt-in or out of using their calculator.

-Line 242/243: Why were 94 participants involved in suspicion percentages for VP3 and 109 participants for VP5? Is this now including participants who used calculator for each question but not nec. Both questions. 

We’ve re-arranged sentences uniformly throughout the results section to make it clearer when we are reporting proportions from all calculator users from one problem or the other versus when we are talking about the sample of people used in the multilevel models who used the calculator on both problems.

We would like to close by thanking the reviewers again for the excellent suggestions. As you can see, they have significantly improved the manuscript. We hope you all agree that the results are exciting and useful to the PloS One readership and that the manuscript now merits publication.

Sincerely,

Mark LaCour

Norma G. Cantù

Tyler Davis

---

## [Decision Letter · Decision Letter 1]

27 Sep 2019

When calculators lie: A demonstration of uncritical calculator usage among college students and factors that improve performance

PONE-D-19-15769R1

Dear Dr. LaCour,

We are pleased to inform you that your manuscript has been judged scientifically suitable for publication and will be formally accepted for publication once it complies with all outstanding technical requirements.

With kind regards,

Valerio Capraro

Academic Editor

PLOS ONE

Additional Editor Comments (optional):

Reviewers' comments:

Reviewer's Responses to Questions

**Comments to the Author**

1. If the authors have adequately addressed your comments raised in a previous round of review and you feel that this manuscript is now acceptable for publication, you may indicate that here to bypass the “Comments to the Author” section, enter your conflict of interest statement in the “Confidential to Editor” section, and submit your "Accept" recommendation.

Reviewer #1: All comments have been addressed

2. Is the manuscript technically sound, and do the data support the conclusions?

Reviewer #1: Yes

3. Has the statistical analysis been performed appropriately and rigorously? 

Reviewer #1: Yes

4. Have the authors made all data underlying the findings in their manuscript fully available?

Reviewer #1: Yes

5. Is the manuscript presented in an intelligible fashion and written in standard English?

Reviewer #1: Yes

6. Review Comments to the Author

Reviewer #1: Thank you for addressing my questions and suggestions.

I am recommending "accept".

Interesting research.

7. PLOS authors have the option to publish the peer review history of their article (what does this mean?). If published, this will include your full peer review and any attached files.

Reviewer #1: No

---

## [Editor Report · Acceptance letter]

4 Oct 2019

PONE-D-19-15769R1 

When calculators lie: A demonstration of uncritical calculator usage among college students and factors that improve performance 

Dear Dr. LaCour:

I am pleased to inform you that your manuscript has been deemed suitable for publication in PLOS ONE. Congratulations! Your manuscript is now with our production department. 

With kind regards,

on behalf of

Dr. Valerio Capraro 

Academic Editor

PLOS ONE